# Cost-Effectiveness of Adjuvanted Quadrivalent Influenza Vaccine for Adults over 65 in France

**DOI:** 10.3390/vaccines12060574

**Published:** 2024-05-24

**Authors:** Marc Paccalin, Gaëtan Gavazzi, Quentin Berkovitch, Henri Leleu, Romain Moreau, Emanuele Ciglia, Nansa Burlet, Joaquin F. Mould-Quevedo

**Affiliations:** 1Geriatrics Department CHU Poitiers, 86000 Poitiers, France; 2CHU Grenoble, 38000 Grenoble, France; 3Public Health Expertise, 75004 Paris, France; 4Seqirus GmbH, Medical Affairs, 81929 Munich, Germany; 5Vifor Pharma, 92042 Paris, France; 6Seqirus USA Inc., Medical Affairs, Summit, NJ 07901, USA

**Keywords:** influenza vaccination, adjuvanted vaccine, cost-effectiveness, France, dynamic model

## Abstract

Background: In France, influenza accounts for an average of over one million consultations with GPs, 20,000 hospitalizations, and 9000 deaths per year, particularly among the over-65s. This study evaluates the cost-effectiveness of the adjuvanted quadrivalent influenza vaccine (aQIV) compared to standard (SD-QIV) and high-dose (HD-QIV) quadrivalent influenza vaccines for individuals aged 65 and older in France. Methods: The age-structured SEIR transmission model, calibrated to simulate a mean influenza season, incorporates a contact matrix to estimate intergroup contact rates. Epidemiological, economic, and utility outcomes are evaluated. Vaccine effectiveness and costs are derived from literature and national insurance data. Quality of life adjustments for influenza attack rates and hospitalizations are applied. Deterministic and probabilistic analyses are also conducted. Results: Compared to SD-QIV, aQIV demonstrates substantial reductions in healthcare utilization and mortality, avoiding 89,485 GP consultations, 2144 hospitalizations, and preventing 1611 deaths. Despite an investment of EUR 110 million, aQIV yields a net saving of EUR 14 million in healthcare spending. Compared to HD-QIV, aQIV saves 62 million euros on vaccination costs. Cost-effectiveness analysis reveals an incremental cost-effectiveness ratio of EUR 7062 per QALY. Conclusions: This study highlights the cost-effectiveness of aQIV versus SD-QIV and HD-QIV, preventing influenza cases, hospitalizations, and deaths.

## 1. Introduction

Influenza is a respiratory infectious disease caused by the influenza virus that includes several strains belonging to type A (H1N1, H3N2 sub-types) and B (Yamagata and Victoria lineages). Influenza epidemic seasons vary from year to year in terms of circulating strains, scale, temporality, and severity. The illness caused by influenza can be mild to severe and presents a risk of death for the infected individual that largely varies with the age of the individual. In France, on average, influenza season leads to more than one million general practitioner (GP) consultations, more than 20,000 hospitalizations, and an estimated 9000 direct and indirect deaths (mean data from the period 2011–2022) [1]. Influenza seasons usually last around 10 weeks with important duration variation across seasons (minimum 8 weeks, maximum 16 weeks) [1].

The attack rate (proportion of people infected) of influenza is challenging to evaluate due to asymptomatic or undiagnosed cases. In unvaccinated individuals, symptomatic and asymptomatic influenza attack rates have been estimated to be 22.5% (95%CI 9.0%, 46.0%) for children and 10.7% (95%CI 4.5%, 23.2%) for adults during a single season [2]. The World Health Organization (WHO) estimates a 5–10% attack rate in adults and 20–30% in children [3]. GP consultations represent the main source of incidence data in France collected by the Surveillance Network (Reseau Sentinelles), but these data are limited to symptomatic diagnosed reported cases. The average incidence rates, estimated from GP consultations, for the last 10 seasons vary from 1000 per 100,000 for people aged 65 years or older up to 8350 per 100,000 for 2–5-year-olds. These rates vary with age due to contact intensity and lack of immunity among the youngest [4].

In addition, influenza severity varies with age. Influenza-related emergency room (ER) visits are more frequent in infants. During the 2021 season, infants represented about 61% of all reported influenza-related ER visits [1]. After infants, adults over 65 represent the largest share of the influenza-related healthcare burden. The immune response of elderly individuals is less effective due to immunosenescence [5,6]. During the 2021 season, adults over 65 represented 42% of hospitalizations, while only representing 20% of the French population [1]. They also represented more than 90% of influenza-attributed deaths in France [1].

Because of the important burden of influenza on the healthcare system, the French Health Authority (HAS) recommends a vaccination campaign for adults aged 65 and over, people at risk of severe or complicated influenza, children aged over 6 months and at risk, and health professionals [7]. The currently available vaccines in France include standard quadrivalent influenza vaccines (SD-QIVs) and the high-dose quadrivalent influenza vaccine (HD-QIV) for adults over 65. The high-dose quadrivalent influenza vaccine (HD-QIV) contains four times the level of hemagglutinin antigen (HA) and has been available in France since 2020. The adjuvanted quadrivalent influenza vaccine (aQIV), which contains the standard dose of HA and MF59 (an oil-in-water emulsion of squalene oil) as an adjuvant that has been designed to produce a greater, broader, and longer immune response [8], has also had marketing authorization in Europe for adults over 65 since 2020 and for adults over 50 since 2023, but has not been reimbursed in France. HD-QIV and aQIV have been shown to have greater vaccine effectiveness compared to SD-QIV for adults over 65 [8,9,10], with HD-QIV and aQIV having similar effectiveness in Real-World-Evidence (RWE) studies. Both aQIV and HD-QIV have been preferentially recommended by National Immunization Technical Advisory Groups (NITAGs) worldwide and the WHO, highlighting their additional benefits compared to SD-QIV [11,12,13]. aQIV has also been shown to be a cost-effective alternative in other European countries compared to the standard-dose vaccine [14,15,16,17,18,19].

The objective of this analysis is to evaluate the cost-effectiveness of aQIV compared to SD-QIV and HD-QIV in the French context for people over 65 years from a societal perspective, excluding indirect costs, as recommended by the HAS [20].

## 2. Materials and Methods

### 2.1. Model Structure

A compartmental dynamic transmission model with a susceptible, exposed, infected, and recovered (SEIR) model (Figure 1) was used to assess the cost-effectiveness of vaccine strategies with separate states and probabilities for the unvaccinated and vaccinated population. The mean duration of influenza exposition was set at 1.8 days, and the mean infectious duration at 0.8 days based on previous publications regardless of the vaccination status [21]. A cycle length of 0.25 days was chosen to best reproduce the propagation of influenza given the short average incubation and infection duration [21]. The model is structured into 5-year age groups, from 0 to 75, with a single 75+ group. It incorporates a French-specific contact matrix [22] to evaluate the rates of contact between these age groups. All model parameters are listed in Table 1.

### 2.2. Model Calibration for the French Context

The calibration of influenza incidence rates in France was based on data from the Sentinelles Network [23], which encompasses data on GP consultations, hospital admissions, and deaths related to influenza. This analysis utilized data from 2011 to 2019 to circumvent the biases introduced by the COVID-19 pandemic. Recognizing that reported cases often exclude asymptomatic individuals and capture only a portion of symptomatic cases, estimates were made regarding the likelihood of exhibiting symptoms and seeking medical consultation if infected with influenza. These estimates aimed to achieve an inferred influenza attack rate of 12.7% for individuals under 18 years old, 4.4% for those aged 18–65, and 7.2% for those over 65, as shown in the meta-analysis by Somes et al. [2]. The derived probabilities were then applied to the number of influenza-related GP consultations to calculate the incidence of influenza across different age groups. Additionally, hospitalization and mortality rates by age group were determined to align with the estimates provided by Santé Publique France [1], taking into consideration the calculated incidence rates of influenza by age group.

The probability of an ER visit was calculated from the hospitalization probability and the rate of hospitalization after ER visit reported by the Géodes Network [24].

### 2.3. Comparators

A current scenario was constructed based on the current vaccine strategies with vaccines currently available in France (SD-QIV) and uptakes for the 2021–2022 seasons (Table 1). Uptakes were obtained from Base De Données Publique Du Médicament [25] given a vaccine coverage by age group of 0.85% for those aged <20 years, 3.45% for 20–64 years, and 51.68% for ≥65 years.

**Table 1 vaccines-12-00574-t001:** Model parameters.

Parameters	Value (95%CI *)	Source
**Natural history**		
Exposed duration	1.8 days [1.44; 2.16]	Nguyen, 2021 [21]
Infectious duration	0.8 days [0.64; 0.96]	Nguyen, 2021 [21]
**Vaccine Coverage**		
<20 years	0.85% [0.68%; 1.01%]	Base de données du médicaments [25]
20–60 years	3.45% [2.76%; 4.14%]	Base de données du médicaments [25]
>60 years	51.68% [41.34%; 62.01%]	Base de données du médicaments [25]
**Probability of a GP Consultation**		Estimated based on average influenza attack rate and reported influenza-related GP consultations [2,23]
<5 years	70% [56.00%; 84.00%]	
5–10 years	50% [40.00%; 60.00%]	
10–15 years	33% [26.40%; 39.60%]	
15–65 years	25% [20.00%; 30.00%]	
≥65 years	17% [13.60%; 20.40%]	
**Probability of hospitalization**		
<5 years	6.2% [4.93%; 7.40%]	Santé Publique France [1]
5–14 years	1.1% [0.91%; 1.37%]	Santé Publique France [1]
15–64 years	1.4% [1.10%; 1.65%]	Santé Publique France [1]
≥65 years	3.6% [2.85%; 4.28%]	Santé Publique France [1]
**Probability of influenza-related death**		
<65 years	0.01% [0.008%; 0.012%]	Santé Publique France [1]
65–74 years	0.22% [0.176%; 0.264%]	Santé Publique France [1]
≥75 years	2.6% [2.08%; 3.12%]	Santé Publique France [1]
**Quality of life**		
General population	0.924–0.756	Szende, 2014 [26]
Disutility of influenza attack (<75, ≥75 years)	0.0087 [0.007%; 0.01%] 0.0074 [0.006%; 0.009%]	Turner, 2003 [27]
Disutility of hospitalization	0.018 [0.014%; 0.022%]	Baguelin, 2015 [28]
**Cost**		
SD-QIV cost	EUR 11.75	French National Insurance [29]
aQIV cost	EUR 23.97 [EUR 19.17; EUR 28.76]	CSL Seqirus France
HD-QIV cost	EUR 30.90	French National Insurance [29]
Vaccine administration cost	EUR 13.94	French National Insurance [30,31,32]
GP consultation cost	EUR 26.78 [EUR 21.42; EUR 32.14]	French National Insurance [29]
Hospitalization cost	EUR 5937.63 [EUR 4750.10; EUR 7125.16]	Efluelda French HTA dossier [33]
ER visit cost	EUR 188.92 [EUR 151.14; EUR 226.70]	French National Insurance [34]
**Base case**		
SD-QIV vaccine effectiveness	40.2% [32.16%; 48.24%]	Belongia, 2016 [35]
HD-QIV relative efficacy vs. SD-QIV	24.2% [9.7%; 36.5%]	DiazGranados, 2014 [9]
**Scenario analyses**		
HD-QIV relative vaccine effectiveness vs. SD-QIV	14.3% [4.2%; 23.3%]	Lee, 2023 [10]
aQIV relative vaccine effectiveness vs. HD-QIV	3.2% [−2.5%; 8.9%]	Coleman, 2021 [8]
aQIV cost	EUR 21.75–EUR 17.98	CSL Seqirus Inc.
HD-QIV and aQIV recommended vaccination age	≥50 years	European Medicines Agency [36]
GP consultation rate for ≥65 years	25–33%	Based on expert opinions
Hospitalization rate for ≥65 years	3.1–16.7%	Pivette, 2020 [37], Fardogrip, 2019 [33]

95%CI *: If the CI is not available, ±20% has been applied to the parameter as lower/upper bound.

This scenario was compared to two scenarios based on using aQIV or HD-QIV instead of SD-QIV in adults over 65 with similar coverage in all three scenarios of 51.68%. In the base case analysis, HD-QIV for 65+ adults was not considered in the current scenario to properly estimate results for aQIV or HD-QIV versus SD-QIV. 

### 2.4. Vaccine Effectiveness

The effectiveness of the SD-QIV vaccine was determined from previous Real-World-Evidence studies, adopting an average effectiveness of 40.2% against influenza infection [35], irrespective of the virus strain or patient age. For HD-QIV, efficacy estimates were derived from its relative vaccine effectiveness (rVE) compared to SD-QIV, based on data from a published comparison regarding laboratory-confirmed cases [9]. An rVE of 24.2% was applied, resulting in an average efficacy of 54.7% for HD-QIV in adults over 65 years. aQIV was assumed to have a similar effectiveness profile, supported by a published meta-analysis [8].

### 2.5. Costs

The study adopted a modified societal perspective excluding indirect costs, as recommended by the HAS [20]. All costs were reported in 2023 Euros (EUR). The assumed unit cost for the aQIV vaccine was EUR 23.97, while for SD-QIV and HD-QIV, the costs were EUR 11.75 and EUR 30.90, respectively, based on the current tariffs [29]. In France, vaccines may be administered by GPs, pharmacists, or nurses. The cost of vaccine administration was calculated based on the average national health insurance (NHI) tariffs for each healthcare professional, amounting to EUR 26.78 for GPs, EUR 7.50 for pharmacists, and EUR 7.56 for nurses [30,31]. The average tariff was calculated on the assumption that each type of healthcare professional administered an equal proportion of vaccines, giving an average tariff of EUR 13.45. Costs for GP consultations and ER visits were also derived from NHI tariffs [32,34]. Hospitalization costs were determined based on the average hospital production cost for diagnosis-related groups (DRGs) associated with influenza, with the cost per hospitalization estimated at EUR 5937.63 in 2023 [33]. 

### 2.6. Utilities

Background utilities per age group for the general French population were obtained from the literature [26]. Utility decrements were applied in case of influenza or hospitalization for the duration of the symptoms. Influenza infection was associated with a disutility of 0.0087 per day for individuals under 75 years and 0.0074 for individuals aged 75 and over [27]. A disutility per hospitalization of 0.018 [28] was considered regardless of the age group.

### 2.7. Analysis

The analysis was based on a single season as an average of the last ten influenza seasons in France to normalize the variability across different seasons. The impact on life expectancy and quality-adjusted life years (QALYs) was assessed over a lifetime horizon, with population data reflective of France. In accordance with HAS guidelines [20], a discount rate of 2.5% was applied to both costs and health outcomes.

To explore various outcomes, multiple alternative scenarios were examined, including different relative vaccine effectiveness (rVE) for aQIV/HD-QIV. This included a scenario where aQIV was assumed to have greater effectiveness than HD-QIV, variations in aQIV/HD-QIV rVE, variations in aQIV pricing, changes in GP consultation and hospitalization rates, and a scenario where adults aged 50 and above would receive vaccination, assuming comparable coverage for individuals aged 50–65 as for those over 65.

Additionally, deterministic sensitivity analyses (DSAs) were performed to assess the influence of key variables on the incremental cost-effectiveness ratio (ICER), examining both the lower and upper limits of these parameters. Probabilistic sensitivity analysis (PSA) was also conducted for these variables, using appropriate distributions (beta distributions for rates, probabilities, and utilities; gamma distributions for costs and normal distribution for durations) to understand their impact on cost-effectiveness outcomes.

## 3. Results

### 3.1. Base Case Analysis

Vaccinating adults over 65 years with aQIV or HD-QIV results in significant reductions in influenza-related morbidity and mortality among this age group (Table 2). This vaccination strategy is projected to decrease the number of GP consultations by 14.05%, hospitalizations by 13.9%, and deaths by 14.12%, translating to approximately 23,967 fewer influenza-related consultations, 854 fewer hospitalizations, and 1594 fewer deaths during an average season. Moreover, the model indicates that vaccinating adults aged 65 and over with aQIV or HD-QIV would also reduce morbidity in other age groups by lowering transmission rates, leading to a 3.2% reduction in GP consultations and a 3.7% reduction in hospitalizations, equivalent to 89,485 GP consultations and 2144 hospitalizations per season.

While these reductions in healthcare consultations and hospitalizations correspond to a decrease in related costs—approximately EUR 9 million for GP consultations and EUR 5.5 million for hospitalizations per season—the switch to aQIV or HD-QIV entails a net increase in vaccination costs, amounting to EUR 110 million for aQIV and EUR 172 million for HD-QIV. Despite the higher upfront costs, the prevention of influenza cases is estimated to yield 14,664 discounted life years (LYs) and 13,514 quality-adjusted life years (QALYs) over individuals’ lifetimes (Table 3). The incremental cost-effectiveness ratio (ICER) compared to SD-QIV is calculated at EUR 7062 per QALY for aQIV and EUR 11,684 for HD-QIV, with HD-QIV considered less cost-effective than aQIV due to a lower price assumption of aQIV compared to HD-QIV.

Deterministic Sensitivity Analysis (DSA) reveals that the price of aQIV is a crucial factor influencing its ICER when compared to SD-QIV (Figure 2). A lower acquisition price (EUR 19.17) significantly reduces the ICER to EUR 3800 per QALY, whereas a higher price (EUR 28.75) increases it to EUR 10,771 per QALY. Price variation has been applied according to a plus or minus 20% basis, as no confidence interval on the price is available. The relative vaccine effectiveness (rVE) and absolute vaccine effectiveness are also key determinants of the ICER, with lower rVE or effectiveness increasing the ICER to EUR 9458 per QALY, and higher values reducing it to EUR 5916 per QALY. Other parameters have a smaller impact on the ICER, with differences under EUR 1000 per QALY. Probabilistic Sensitivity Analysis (PSA) indicates that in 90% of simulations, the ICER remains below EUR 15,000 per QALY, and in 95% of cases, it stays below EUR 21,000 per QALY (Figure 3 and Figure 4).

### 3.2. Scenario Analysis

The scenario results are presented in Table 4. When considering a relative vaccine effectiveness (rVE) of aQIV versus HD-QIV at 3.2% [8], the ICER decreases to EUR 6304 per QALY, positioning aQIV as the dominant strategy. Opting for a more conservative rVE for aQIV/HD-QIV versus SD-QIV at 14.3% results in an increase in the ICER to EUR 12,525 per QALY. Adjusting the price of aQIV to EUR 21.75 and EUR 17.98 leads to a reduction in ICERs to EUR 5469 per QALY and EUR 3069 per QALY, respectively. Implementing vaccination with aQIV for individuals over 50 years with a similar vaccine coverage emerges as a dominant strategy compared to SD-QIV, yielding savings of EUR 7 million and enhancing overall utility. Altering the GP consultation rate for those aged 65 and over to 25% and 33% leads to a decrease in ICER to EUR 6867 per QALY and EUR 6671 per QALY, respectively. Utilizing alternative hospitalization rates (3.1% for those aged 65+) results in an ICER of EUR 7117 per QALY, while adopting alternative hospitalization rate (16.7% for those aged 65+) reduces the ICER to EUR 5536 per QALY.

## 4. Discussion

This analysis demonstrates the potential benefits and cost-effectiveness of implementing the adjuvanted quadrivalent influenza vaccine (aQIV) and high-dose quadrivalent influenza vaccine (HD-QIV) in adults over 65 years in the French healthcare context. These findings align with and extend upon existing literature that emphasizes the importance of targeted influenza vaccination strategies to mitigate the healthcare burden associated with influenza infections, particularly in older adults and high-risk groups [38].

The limitations of the analysis include the reliance on historical data for model calibration, which may not fully capture future variations in influenza season severity or vaccine effectiveness due to evolving virus strains. Moreover, relying on GP consultations, which likely underestimate the influenza attack rate, along with the challenges in accurately attributing hospitalizations and deaths to influenza due to coding practices, injects a degree of uncertainty into the findings. Exploring this uncertainty through alternative data sources, especially concerning adults aged 65 and over, has nonetheless led to comparable ICERs. Finally, the focus on direct medical costs excludes the indirect costs associated with lost productivity or long-term disability from severe influenza cases. Future research should consider these factors to provide a more comprehensive assessment of the cost-effectiveness of influenza vaccination strategies, which would likely lead to even lower ICERs.

The observed reduction in influenza-related morbidity and mortality, seen in the decrease of 90,000 GP consultations, 2100 hospitalizations, and 1600 deaths per year, underscores the effectiveness of aQIV and HD-QIV in reducing the transmission and impact of influenza. This is consistent with previous studies that have highlighted the superior effectiveness of these vaccines in older adults compared to standard-dose vaccines (SD-QIVs) [8,9,39,40]. Furthermore, the use of a dynamic model indicates that using more effective vaccines in adults aged 65 and over would also benefit the other age groups through reduced transmission, further increasing the benefit of using aQIV and HD-QIV. Over individuals’ lifetimes, the morbidity and mortality reduction translates into a substantial increase in LY and QALY. When considering the additional cost associated with aQIV compared to SD-QIV, the ICER obtained for aQIV is EUR 7062 per QALY versus SD-QIV, demonstrating a strong cost-effective strategy at a threshold of EUR 30,000 per QALY. Sensitivity analysis confirms the robustness of our findings and echoes findings from similar studies showing that cost-effectiveness outcomes for influenza vaccination are sensitive to vaccine pricing, vaccination coverage, and relative vaccine effectiveness (rVE) [17,38,41,42]. Both in sensitivity analysis and when using alternative sources for some parameters, the ICERs consistently remain below EUR 11,000 per QALY, demonstrating the robustness of the results.

The incremental cost-effectiveness ratios (ICERs) presented in the base case and scenario analyses are informative for healthcare policymakers, indicating that aQIV and HD-QIV could be cost-effective options under certain conditions, especially when considering the broader implications for public health and healthcare systems. This complements previous findings where aQIV and HD-QIV have been shown to have similar ICERs in similar populations, including an ICER of EUR 4527 per QALY in Italy [38], EUR 2660 per QALY in Argentina [41], EUR 6694 per QALY in Spain [17], and EUR 10,170 per QALY [19] or EUR 4365 per QALY in the US [42]. It has also been demonstrated that in other locations, aQIV can be cost-saving compared to HD-QIV [14,16,43].

The scenario including vaccination for adults aged 50 with a 58% coverage yielded better results than previously reported [43]. However, this is likely because, for adults aged between 50 and 65, the coverage considered in the current scenario with SD-QIV was 3.45%, effectively comparing aQIV to no vaccine. However, this result is informative as it suggests that implementing vaccination for adults aged between 50 and 64 with aQIV, for which aQIV just received EMA approval, would be cost-saving overall compared to the current scenario, financing the increase in costs associated with switching to aQIV for adults over 65.

The number of deaths estimated by the model for an average season is 11,981 in the scenario with SD-QIV and 10,370 with aQIV or HD-QIV. These estimates are higher than the 9000 average number of deaths reported by Santé Publique France between 2011 and 2020 [1]. However, this average includes an abnormally low figure of 702 deaths during the 2013–2014 season. With the input from the experts consulted during this project, we decided not to consider this amount in our mortality rate calculation. Excluding it, the average number of deaths increases up to 10,114 over the same period, which is closer to the model estimations.

## 5. Conclusions

In conclusion, this analysis supports the adoption of aQIV and HD-QIV for adults aged over 65 in France as a potentially cost-effective strategy to reduce the burden of influenza. Adjuvanted and high-dose vaccines present, respectively, ICERs of EUR 7062 and EUR 11,684 per QALY compared to the standard-dose vaccine. They also prevent 3.7% of hospitalizations and 13.4% of deaths attributed to influenza. Inherent limitations of this analysis include the representativeness of influenza data, which tend to underestimate the true burden of the disease. However, the findings remain consistent with those of other comparable studies and underscore the potential of novel vaccine modalities. It also highlights the importance of flexible vaccination strategies that can be adapted based on evolving evidence and changing healthcare landscapes.

## Figures and Tables

**Figure 1 vaccines-12-00574-f001:**
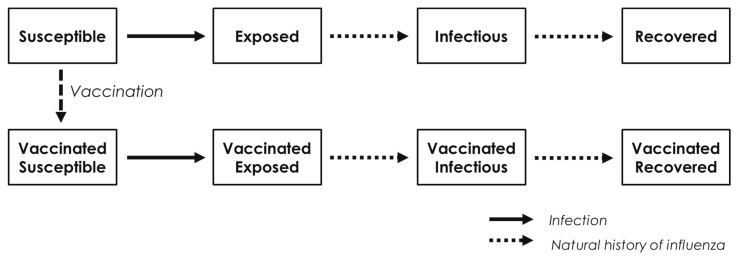
SEIR (susceptible, exposed, infectious, recovered) model structure used for the cost-effectiveness analysis. Natural history of influenza refers to the evolution of the disease from infection to cure or death.

**Figure 2 vaccines-12-00574-f002:**
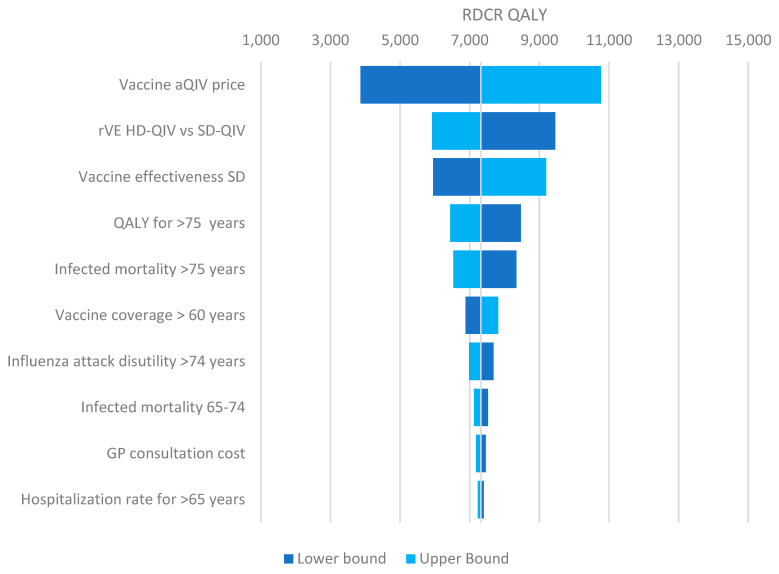
Tornado diagram: impact of the deterministic sensitivity analysis on the incremental cost per quality-adjusted life-year (ICER EUR /QALY) associated with vaccinating adults 65 and over with aQIV instead of SD-QIV.

**Figure 3 vaccines-12-00574-f003:**
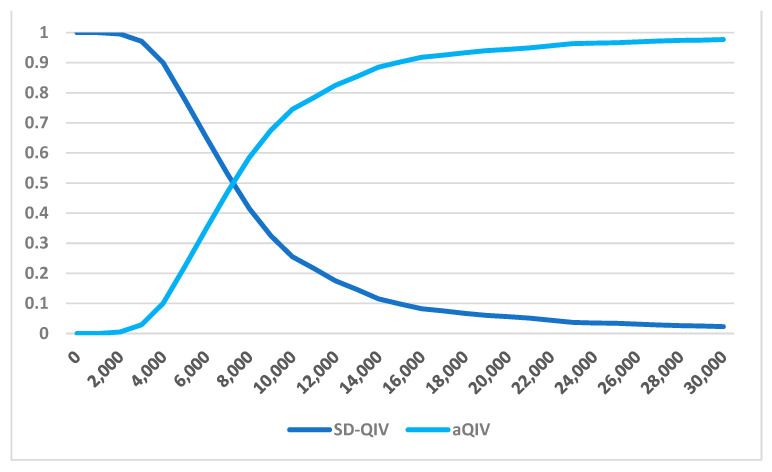
Cost-effectiveness acceptability curve (SD-QIV vs. aQIV).

**Figure 4 vaccines-12-00574-f004:**
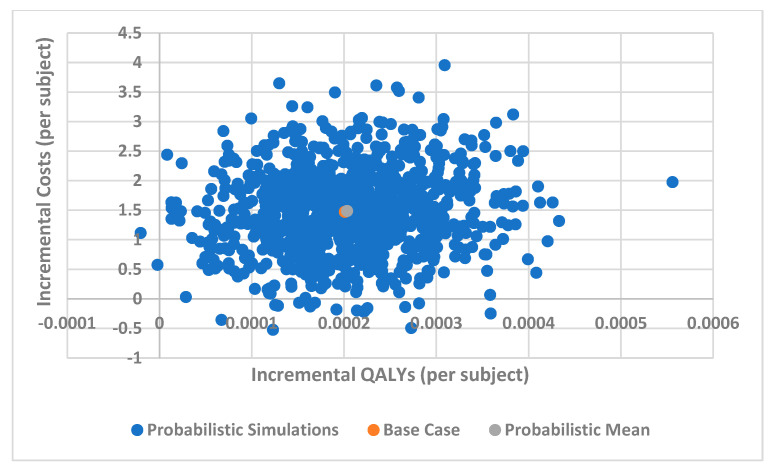
Cost-effectiveness scatter plot (SD-QIV vs. aQIV).

**Table 2 vaccines-12-00574-t002:** Base case results: Comparison of the effectiveness and cost for each vaccine used in adults aged over 65 years.

	S0: Vaccination of 65+ with SD-QIV Only	S1: Vaccination of 65+ with aQIV Only	S2: Vaccination of 65+ with HD-QIV Only
Vaccinations <65 years >65 years	10,244,491 3,292,292 6,952,199	10,244,491 3,292,292 6,952,199	10,244,491 3,292,292 6,952,199
GP consultations <65 years >65 years	2,815,814 2,645,225 167,436	2,726,329 (−3.2%) 2,579,707 (−2.3%) 146,622 (−14.0%)	2,726,329 (−3.2%) 2,579,707 (−2.3%) 146,622 (−14.0%)
Hospitalization <65 years >65 years	57,513 51,432 6081	55,370 (−3.7%) 50,143 (−2.24%) 5227 (−14.0%)	55,370 (−3.7%) 50,143 (−2.24%) 5227 (−14.0%)
Deaths <65 years >65 years	11,981 696 11,284	10,370 (−13.4%) 679 (−0.14%) 9691 (−14.1%)	10,370 (−13.4%) 679 (−0.14%) 9691 (−14.1%)
Acquisition cost (EUR)	130,860,961	240,962,190 (+84%)	303,422,952 (+132%)
Administration cost (EUR)	137,788,397	137,788,397 (0%)	137,788,397 (0%)
GP consultation cost (EUR)	246,202,735	238,246,853 (−4%)	238,246,853 (−4%)
Hospitalization cost (EUR)	39,225,719	33,714,775 (−14%)	33,714,775 (−14%)
Total cost (EUR)	555,252,778	650,712,216 (+17%)	713,172,978 (+28%)

**Table 3 vaccines-12-00574-t003:** Base case results: average discounted life years (LYs), quality-adjusted life years (QALYs), and costs for each vaccine used in adults aged over 65 years.

	Cost	QALY	LY	ICER (EUR/QALY)	ICER (EUR/LY)
S0: vaccination of 65+ with SD-QIV only	EUR 555,265,389	1,365,910,126	1,665,735,095	-	-
S1: vaccination of 65+ with aQIV only	EUR 650,712,216	1,365,923,640	1,665,749,760	-	-
S2: vaccination of 65+ with HD-QIV only	EUR 713,172,978	1,365,923,640	1,665,749,760		
Δ S1 vs. S0	EUR 95,446,442	13,514	14,664	EUR 7062 per QALY	EUR 6508 per LY
Δ S2 vs. S0	EUR 157,920,200	13,514	14,664	EUR 11,684 QALY	EUR 10,768 LY
Δ S1 vs. S2	EUR −62,460,763	-	-	Dominant	Dominant

**Table 4 vaccines-12-00574-t004:** Scenario analysis results for vaccinating adults aged 65 and over with aQIV or HD-QIV instead of SD-QIV.

Scenario	GP Consultations Avoided	Hospitalizations Avoided	Deaths Avoided	ICER (EUR/QALY)
Base case				
S0 vs. S1	−89,485	−2144	−1611	EUR 7062 per QALY
rVE a-QIV vs. HD-QIV: 3.2%				
S0 vs. S1	−98,307	−2354	−1766	EUR 6304 per QALY
S1 vs. S2	−8822	−210	−155	EUR −49,112 per QALY
rVE HD-QIV vs. SD-QIV: 14.3%				
S0 vs. S1	−53,259	−1277	−966	EUR 12,525 per QALY
Fluad Price EUR 21.57				
S0 vs. S1	−89,485	−2144	−1611	EUR 5469 per QALY
Fluad Price EUR 17.98				
S0 vs. S1	−89,485	−2144	−1611	EUR 3069 per QALY
Vaccination of patient 50+ with HD-vaccine				
S0 vs. S1	−158,172	−3505	−1934	EUR 4891 per QALY
GP consultation rate for 65+: 25%				
S0 vs. S1	−100,762	−2545	−1611	EUR 6867 per QALY
GP consultation rate for 65+: 33%				
S0 vs. S1	−112,041	−2947	−1611	EUR 6671 per QALY
Pivette et al. hospitalization rate: 3.1%				
S0 vs. S1	−89,485	−2032	−1611	EUR 7116 per QALY
Fardogrip hospitalization rate: 16.7%				
S0 vs. S1	−89,485	−5291	−1611	EUR 5536 per QALY

## Data Availability

Data are contained within the article.

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
