# Peer review of "Cost-Effectiveness of Adjuvanted Quadrivalent Influenza Vaccine for Adults over 65 in France"

_vaccines, 2024, doi:10.3390/vaccines12060574_

Round 1

Reviewer 1 Report

Comments and Suggestions for Authors

The manuscript describes a study evaluating the cost-effectiveness of using an adjuvanted influenza vaccine, considering the French reality of public health. The authors concluded, based on the analysis of a series of hospitalization cost data and consultations on the effectiveness of this type of vaccine. As for the manuscript itself, I can say that, overall, it is well-written, with clear objectives, in presenting the data, and data-supported conclusions. However, I have many reservations about this type of study.

I understand the need for this type of analysis, considering health expenditures in general, both for populations and governments (where there are public vaccination campaigns). But for me, there is no way to quantify death, for example. It is not possible to place this variable in an equation. Suppose we only consider hospitalization costs and the potential loss of work from economically active members of a population. In that case, that's fine, but I am very hesitant when death and quality of life are included in the equation.

This was mentioned in passing in the last paragraph of the discussion, about the limitation of the study. But I think this limitation should have been highlighted from the beginning of the work. In conclusion, does this reviewer understand that the manuscript should be published? Yes, its publication may be interesting because it highlights the importance of a vaccine formulation. However, I suggest that the discussion consider the limitations of the work from the beginning (not just in the last paragraph).

Author Response

Dear Reviewer,

I sincerely appreciate your feedback on our article. A perspective such as yours is invaluable for ensuring the accuracy and clarity of our arguments, as well as the overall quality of our work.

I understand your concerns regarding the inclusion of mortality and quality of life considerations in this study. However, in conducting a health economic analysis to assess the effectiveness of a public health strategy, the impact on patients' quality of life is an essential variable. Collected using scientific methods, quality of life data allows for an objective assessment of patients' experiences during illness, and quantifies the well-being preserved when an individual does not contract a disease due to medication or vaccination. Furthermore, quantifying deaths is imperative in analyzing the epidemiological consequences of a lethal disease such as influenza. While both types of data have inherent limitations, they are accounted for in studies with designs similar to ours.

I concur that these limitations should neither be minimized nor concealed. Accordingly, we will relocate the paragraph detailing them to earlier in the discussion section to address this issue sooner. The corresponding paragraph was moved from the line 299 to the line 246.

Thank you once again for your valuable input.

Reviewer 2 Report

Comments and Suggestions for Authors

Dear Authors!

Thank you for opprtunity to review your manuscript.

Vaccination is a known high effective tool for saving patients' lives and decrease the burden of the infection on the healthcare system and economy in common. The manuscript confirms with modern analysis the real cost-effectiveness of the vaccination against flu in elder people. 

The manuscript is actual and actuality of the study is high due to save the lives and economic efficacy.

The goals are clear

The Methods are clear and described in details

The results corresponds to the methods and have tales and figures, confirming the study results

The Discussion contains all necessary literature and Authirs compared their results with previously published data

The discussion contains Limitation subsection in the end of the discussion section. I suggest to organize the specific subsection with Heading "Limitations"

The conclusion is interfered the study results and underlines them

I have no any principal comments

Author Response

Dear Reviewer,

Thank you for taking the time to review our manuscript on the medico-economic analysis of adjuvanted vaccine in France for elderly people. We appreciate your time and effort in providing feedback. Your contribution is invaluable to the refinement and quality of our research. Should you have any further comments or suggestions, please do not hesitate to reach out.

Best regards,

Quentin Berkovitch

Public Health Expertise

Reviewer 3 Report

Comments and Suggestions for Authors

The survey paper, “Cost-effectiveness etc..”, by Marc Paccalin et al. is a statistical analysis of the effectivity of the currently used Flu vaccines in France, which suggests that the adjuvanted quadrivalent vaccine is the winner, particularly in the > age group. The data sampling, stratification of the results, statistical analyses are all superb, and the conclusions are convincing.

I have a few minor comments / suggestions:

1) The lower immunity of the above-65 age group is generally referred to as “immune senescence”. This term should be used in the first appropriate place, for example Line 56.

2) In Line 46, explain the term “attack rate”.

3) Figure 1 needs a better clarification of “Infection” versus “Natural history of influenza”.

4) The Conclusions section is well-written and concise, but slightly vague. I suggest that this section should include a few other observations from different aspects of the study. An even better suggestion is to state the conclusion for each subsection at the end of the subsection, which is currently absent.

Minor English corrections will help, such as “contacts intensity” (Line 51), “an ER visits” and “probability..were”  (Line 113).

Comments on the Quality of English Language

Minor, only in a few places. A couple of examples given in my Comments.

Author Response

For research article

Response to Reviewer 2 Comments

1. Summary

Thank you for taking the time to review our manuscript on the medico-economic analysis of adjuvanted vaccine in France for elderly people. We appreciate your time and effort in providing feedback. Your contribution is invaluable to the refinement and quality of our research. All parts of the manuscript modified in accordance with your comments will be highlighted in blue.

2. Questions for General Evaluation

Reviewer’s Evaluation

Response and Revisions

Does the introduction provide sufficient background and include all relevant references?

Yes

Are all the cited references relevant to the research?

Yes

Is the research design appropriate?

Yes

Are the methods adequately described?

Yes

Are the results clearly presented?

Can be improved

Are the conclusions supported by the results?

Yes

3. Point-by-point response to Comments and Suggestions for Authors

Comments 1: The lower immunity of the above-65 age group is generally referred to as “immune senescence”. This term should be used in the first appropriate place, for example Line 56.

Response 1:

Thank you for this precision. A new sentence containing the immunosenescence term has been integrated line 57, with two references underlining the phenomenon for influenza. This sentence is highlighted in blue.

Line 57: “The immune response of the elderly is less effective due to immunosenescence[5,6].”

[5]: Haq K, McElhaney JE. Immunosenescence: influenza vaccination and the elderly. Curr Opin Immunol 2014;29:38–42. https://doi.org/10.1016/j.coi.2014.03.008.

[6]: McElhaney JE, Verschoor CP, Andrew MK, Haynes L, Kuchel GA, Pawelec G. The immune response to influenza in older humans: beyond immune senescence. Immun Ageing 2020;17:10. https://doi.org/10.1186/s12979-020-00181-1.

Comments 2: In Line 46, explain the term “attack rate”.

Response 2:

We agree on the precision about the definition of the attack rate. Thank you for pointing it out. We added a precision between brackets.

Line 43: “The attack rate (proportion of people infected)”.

Comments 3: Figure 1 needs a better clarification of “Infection” versus “Natural history of influenza”.

Response 3:

Thank you for your comments. We clarified the term “Natural history of influenza” with a short sentence.

Line 98: “Natural History of influenza refers to the evolution of the disease from infection to cure or death.”

Comments 4: The Conclusions section is well-written and concise, but slightly vague. I suggest that this section should include a few other observations from different aspects of the study. An even better suggestion is to state the conclusion for each subsection at the end of the subsection, which is currently absent.

Response 4:

Thank you for your remarks. We specified the conclusions with a group of sentences resuming cost-effectiveness and epidemiological results, some limitations regarding the data and the consistency of the results with similar studies.

Line 302: “Adjuvanted and high-dose vaccines presents respectively an ICER of €7,062 and €11,684 per QALY compared to standard-dose vaccine. They also prevent 3.7% of hospitalizations and 13.4% of deaths attributed to influenza. Inherent limitations of this analysis include the representativeness of influenza data, which tends to underestimate the true burden of the disease. However, the findings remain consistent with those of other comparable studies and underscore the potential of novel vaccine modalities.”

4. Response to Comments on the Quality of English Language

Point 1: “contacts intensity” (Line 51)

Response 1: “contact intensity”

Point 2: ”ER visits” (Line 113)

Response 2: “ER visit”

Point 3: “probability…were” (Line 113)

Response 3: “probability…was”